# A Proposed Framework on the Affective Design of Eco-Product Labels

**Alma Maria Jennifer Gutierrez** * **, Anthony Shun Fung Chiu** and **Rosemary Seva**

Industrial Engineering Department, De La Salle University, Manila 1004, Philippines;
anthony.chiu@dlsu.edu.ph (A.S.F.C.); rosemary.seva@dlsu.edu.ph (R.S.)
* Correspondence: alma.gutierrez@dlsu.edu.ph

**Abstract:** There was a shift in sustainability consumption in the last decade that stimulated new strategies for ecological friendly industries and new product innovations. Environmental labeling is a marketing technique used to inform consumers that a company has employed a process to protect the environment. However, uncertainty remains concerning how eco-labels influence consumers. Buying green products can elicit emotion in consumers. When consumers buy eco-products, they feel that they are helping save the environment. Products provide certain emotional benefits and therefore affect mood and behavior. This study aims to examine how consumers who differ in environmental attitudes respond to eco-labels. Aside from this, it wants to determine the intensity and type of emotions elicited by these kinds of products based on a certain set of pre-purchase emotions. These emotions are still unknown. Also, it proposes the Green Emotion Model (GEM) 2.0 that correlates environmental attitudes, visual attention towards these eco-labels, emotion and the desirability of purchasing a product. This framework proposes that the environmental attitudes and awareness of consumers are crucial for them to look for this eco-label on a product. These environmental labels should be able to capture the attention of consumers and thus will provoke positive emotions and lead to the purchase of an eco-product.

**Keywords:** affect; eco-labels; pre-purchase

## 1. Introduction

### 1.1. Background of the Study

Green marketing continues to grow due to the shift in consumption of eco products [1]. The industry was estimated to be worth over $200 billion in 2006 [2]. "There is an increase in the environmental conscience of consumers which results in the increased demand for green products an observation that several companies have taken advantage of by offering these products and services" [3–7]. Eco products are products that operates within a sustainable way over the entire product life cycle thus protecting the environment [8–11]

Labels on products provide an information for a consumer to make choices. Sustainability labels is a marketing technique for informing consumers that a company has engaged in a process to protect the environment. In advertising these eco-products, it has to publicize the message either through symbols or claims on labels as to the type of ecological benefits that the product has to offer. There are different ways in which marketing can show the environmental benefits of products through product claims such as the product being "eco-friendly", "environmentally safe", "recyclable", "biodegradable" and "ozone-friendly" [12]. If consumers understand the importance of these environmental characteristics, then they may be willing to pay a higher price for the product.



Many consumers have been interested and are willing to pay a premium for green characteristics [13–17] and that they do purchase such products [18–20]. Environmental labels are valued as a key to achieving a purchase outcome and therefore considered as an important consideration that will influence consumers in their purchasing patterns [21,22]. Green labels act as a guide for consumers to choose environmentally friendly products. Companies use this to characterize their products, position them and communicate an environmentally friendly message [23].

Buying green products can elicit emotions in consumers. They trigger a unique set of emotional responses. These emotions are normally positive reaction, making consumers interested to these products and choose to buy them [24]. The altruistic attitude on patronizing eco products cannot be denied [25]. Consumers therefore can feel a sense of fulfillment by contributing to the improvement of the environment.

Since green products are pricey than their counterparts in the market, [26] proposed a set of procedure based on either functional brand attributes or emotional benefits. They explored and tested the magnitude of green marketing and its effects on brand attitude. Results showed an overall positive impression of green brand positioning on brand attitude. The highest outcomes were achieved through a green marketing segmentation that combined functional attributes with emotional benefits. This study, however, did not identify the attributes of the products that can contribute to positive emotions toward the environment. The results are applicable for marketing the product and not for designing the product to elicit positive emotions from the buyers.

The preliminary literature review revealed that environmental information characteristics that influences affective responsive had not yet been undertaken in previous researches. Aside from this, the emotional attachment in the pre-purchase context of green products had not yet been explored.

*1.2. Objectives of the Study*

This research aims to:

1.  Identify the ecological information of consumer products eliciting intense emotions that will eventually lead consumers to purchase the product.
2.  Determine the emotions engendered by consumer eco-products in the pre-purchase stage. Most decisions that people make are led by their emotions. Consumers' judgment of products does not rely only on brand, quality or price but also on the affect experienced during purchase and consumption [27].
3.  Identify the eco-information that influences pre-purchase affect. This involves defining the ecological information that is capable of eliciting affect and thus influencing consumers' purchase intention.
4.  Predict purchase intention from emotions experienced in the pre-purchase stage using the proposed Green Emotion Model (GEM) 2.0. This involves predicting how product attribute for purchase may engender affect.

## 2. Hypotheses and Model Development

*2.1. Hypotheses and Research Problems*

**Hypothesis 1 (H1):** *Consumers interest to eco-products in the pre-purchase stage will result to a distinct model of affective responses.*

Establishing the distinct model of emotional reactions to consumer eco-products in the pre-purchase stage is critical for measuring the influence of product attributes. Knowing the ecological information that will trigger positive emotions in the consumer leads to purchase intention. Previous researches that dealt with the influence of affect were too varied and have a wide context of application.

**Hypothesis 2 (H2):** *Intense positive emotion of customers on green products influence purchase intention.*

When a consumer feels an intense positive responses when inspecting a product, the favorable outcome will motivate the consumers to buy the product that causes this emotion. A strong positive emotion will directly impact the behavior on the purchase decisions. However, no studies were conducted on the correlation between positive emotion and purchase intention for consumer-based eco-products. No studies were conducted on what ecological information triggers emotions or whether an emotion is sufficient to form a purchase intention.

**Hypothesis 3 (H3):** *A positive environmental attitude strengthens the relationship between ecological attributes and emotion.*

Ninety percent of American consumers claim that they care about the environmental consequences of their purchases and 75% stated that they include environmental considerations in their shopping decisions or consider themselves as being eco-conscious [28]. Capitalizing on these environmental attitudes, manufacturers can determine whether ecological information results in positive emotions for consumers and encourages purchases of eco-products.

*2.2. Green Emotion Model (GEM) 2.0*

The Green Emotion Model (GEM) 2.0 framework suggests that consumers' environmental attitudes influence their behavior by encouraging them to engage with eco-labeling messages in a way that elicit positive emotions. This engagement in turn increases consumers' desire to purchase eco-products. This framework is outlined in Figure 1. This framework would like to investigate how consumers who differ in terms of their environmental attitudes respond to eco-labels. It also tries to manipulate eco-labels in terms of their locations on the packaging, their size and their color as factors that capture consumers' attention. It also explores Type 1 and Type 2 self-declaration claims made by companies and determines whether they are understandable to consumers. Lastly, it probes the emotions provoked by these products because most choices that people make are ruled not only by reason but also by emotions. Consumers acquire certain products because their emotions are triggered during the purchasing process.

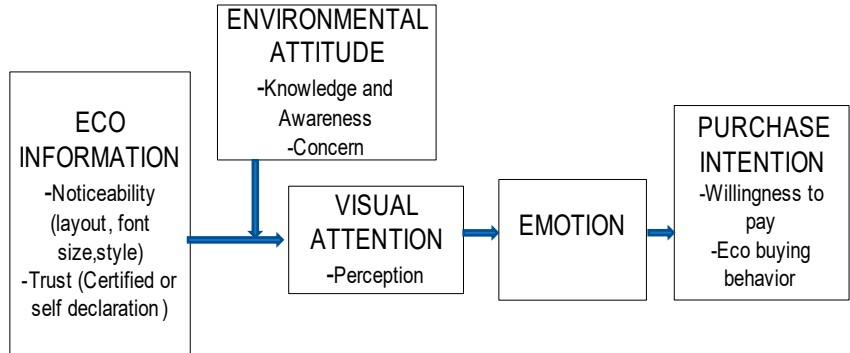

**Figure 1.** The Green Emotion Model (GEM) 2.0.

2.2.1. Eco-Information Attributes

Labels are designed to be seen by consumers, including any eco-information attributes placed on a product's packaging. Product designers should therefore ensure that consumers' attention is captured at the shopping stage. The layout or location of this information, as well as the font size and type of font used when presenting the information to consumers, are critical to visual perception. Research conducted about consumers' perceptions of the Energy Star Label suggested revision of the label to increase clarity. The new logo included the words "Energy Star" on each label and uses a strong clear

blue color [29]. This example shows that it is important for the readability of eco-information attributes to be considered when placing eco-information claims on consumer products.

Second, the trust that a consumer places on the information provided as well as the source of that information is also a major issue. International Organization for Standardization (ISO) 14020 series deals with aspects of environmental labels and declarations. It developed and classified different types of labeling based on their type number. The two main types are already in existence. Third-party verification or Type 1 labels are the classic eco-labeling methods, which award a mark or logo based on set criteria. These labels include Green Seal, Energy Star, Scientific Certification Systems (SCS), the Forest Stewardship Council (FSC), the Leadership in Energy and Environmental Design (LEED), Chlorine Free Products Association, Energy Star, Veriflora, Fair Trade Certified, Leaping Bunny, Marine Stewardship Council and Green E. [30] Type 2 labels are claims made by manufacturers and businesses, meaning they are self-declared environmental claims. The question remains whether these marks or logos are recognizable and understood by the consumers. Consumers recognized the difficulties in distinguishing the differences among the six-plus eco-labels for coffee [29]. Studies have shown that eco-labels endorsed by government agencies like the Energy Star, which is supported by the U.S. Environmental Protection Agency and the Department of Energy, is perceived as a credible standard.

### 2.2.2. Environmental Attitude

Environmental attitude is characterized as a complex combination of personal experiences, cultural norms and values, which underlines opinions towards an environmental issue [31]. An environmental attitude originates from three factors: environmental worldview, concern and commitment [31].

Environmental knowledge is important for understanding environmental problems and issues. Aside from building awareness and a positive attitude towards the environment, this knowledge also enables consumers to engage in a variety of experiences [31]. Environmental knowledge also triggers concern and attention among individuals that can lead to a willingness to act to alleviate environmental problems [31]. A consumer with a positive environmental attitude will look for the eco-information attributes (eco-label) on the products they buy. Consumers may do this in response to environmental problems. Several studies have shown that increasing the environmental awareness of customers has inspired them to be more motivated towards purchasing the goods of environmentally friendly brands [32] Capitalizing on these environmental attitudes, manufacturers can determine whether eco-information attributes can elicit positive emotions in consumers and result in consumer purchases of eco-products.

### 2.2.3. Visual Attention

People rely on visual information to make decisions such as making purchases. The typical consumer scans the items on each shelf to look for the brand s/he has in mind or inspect each product before deciding to purchase. The study focused on the gaze behavior of the consumer while searching for a product in a supermarket [33]. They used an eye tracker to record eye movements in the actual environment. Eye-tracking is a procedure where a person's eye movements are measured to determine where he/she is looking at any given time and the progression in which their eyes are changing from one point to another. In the context of a supermarket, tracking consumer eye movements can help to understand how a consumer searches through a display of items on supermarket shelves. It can help explain the decision process and compare it to their search process. The eye-tracking process showed the differences between a decision-making task and a search task. The result of the study affirms that the evaluation stage of a decision task contains more re-dwells than a comparable search task does.

A similar exploratory study examined the relationship between abundant in-store stimuli and limited human perceptual capacity [34]. They studied the impact of package design characteristics on visual attention. Two eye-tracking experiments were conducted in an actual supermarket and a laboratory setting. Based on the results obtained, consumers have divided visual attention during grocery shopping. Their visual attention is dominated at the same time by the shelf displays.

Design features like contour/shape, contrast and ratio can be advantageous for drawing the visual attention of consumers towards products.

### 2.2.4. Emotion

Consumers nowadays are more demanding because they do not just want a product that is functional and usable, but also pleasurable [35]. It is the interaction of people and products that matters in the consumption stage.

Most of the brain activity is emotional and not cognitive according to George Loewenstein [36]. Emotion influences our daily lives. "Emotion dominates people's decision making, commands attention and enhances some memories while minimizing others" [35] "Emotion is the energy that drives and directs our attention" [35]. Emotions and other affective state and personality traits influence every aspect of our interaction with brands and products [35]. Positive emotions evoked by a product may lead to a person's purchase decision.

"Emotional value is the observed utility derived from a substitute capacity to arouse feelings or affective states" [37]. It tries to estimate consumer emotions toward green products. Goods and services are influenced with emotional responses [38]. The influence of rational and emotional factors comes into play when a consumer is interested to a particular product or service, however emotions are crucial in every purchase decision [39]. "A similar study conducted revealed emotional value, conditional value and epistemic value all positively influence respondent choice behavior towards green products" [38]. As a result, consumers often attach higher emotional value to green products. Consumers who prefers going green feels that they are helping to protect the environment and have the potential to experience positive feelings when purchasing green products because they see these purchases being good for them and for society at large.

### 2.2.5. Purchase Intention

Numerous studies have shown an ever-increasing interest for eco-labeled or green products that consumers are willing to pay (WTP) a higher price [13,14,16,17,40–43]. "Consumers are expected to double their spending on "green products", as they quickly reached up to $500 billion of spending in 2009" [44]. If consumers are willing to purchase these eco products, companies will be influenced to adhere to environmental legislation and regulations and therefore protect the environment. According to James Ludwig, the director of Steelcase, companies that do not consider the environment in mind when designing products are essentially sleepwalking and in for a rude awakening because "protecting the environment is becoming more of a current driver of the market" [45].

Consumers purchasing patterns for green products revealed their willingness to pay a premium price. It is reasonable to say that when a consumer experiences positive emotion, they will be inclined towards favorable responses such as buying the product that triggers these emotions. "Purchases of organic foods are motivated both by expected positive consequences for the self and others" [46]. Previous studies also showed that consumers sometimes relate the altruistic attitude and feelings of responsibility for the well-being of one's family with organic food purchasing decisions [16,47,48]

This study will provide information on the importance of studying the design variables of eco-labels such as the size, location/layout and color, as well as whether consumers trust these eco-labels or not. An eco-label is supposed to inform consumers about what a company has done to protect the environment, so it is important for consumers to notice them. This study will offer knowledge of the effectiveness of signaling ecological attributes or eco-labels to consumers. This will help companies in developing and designing eco-labels that are appealing, noticeable and favored by consumers.

### 2.3. Significance of the Study

This framework proposes that the environmental attitudes and awareness of consumers are crucial for them to decide to look for eco-labels on a product. These environmental labels should be able to

capture the attention of consumers and thus will provoke positive emotions. Previous studies have established the effect of aesthetics, usability and functionality on the experience of positive emotions, but none have tried to link environmental attributes to emotions that lead to strong buying intentions. This study will help marketers of environmentally friendly products to establish strategies on how to communicate these environmental attributes that will influence purchase intentions.

Currently, majority of organizations do not offer sustainable products because there is still uncertainty whether consumers are willing to pay more for the "environmentally-friendly" feature. This study hopes to entice organizations, as it will try to establish the affective attachment of consumers to green products.

To our knowledge, this is the first attempt at showing the relationship of eco-information, environmental attitudes, visual attention, emotion and purchase intention into one model. Previous studies have focus on the eco-information and environmental attitude of consumers alone. These studies believe that the success of eco-information depends on the environmental awareness of consumers. The premise of this study is that educated individuals are more likely to look for eco-information/eco-labels than those that are less educated or are environmentally aware. No studies have tackled the noticeability of eco-information that can trigger positive emotions and significantly impact consumer perceptions and purchases of eco-friendly products. Other studies only focus on the importance of emotion on the purchase intention of consumers. However, no studies have linked all of the five factors into one. This framework can be used by manufacturers when designing the packaging of their products. If they want to convey that their product is eco-friendly, then their eco-labels should be noticeable, and the information presented should be understandable to the consumer. Aside from this, the credibility of the eco-label will increase positive eco-behavior towards the brand from consumers. Lastly, if consumers understand and trust eco-labels, then this will trigger positive emotions and therefore lead to the purchase of the products with eco-labels.

## 3. Data and Methods

Data gathering was divided into three parts: The first part involved identifying pre-purchase affect while buying eco-products through field and online surveys. The second part involved seeking to determine the awareness of people towards eco-labels on consumer products through an online survey.

The third part determined the emotional structure of the pre-purchase affect so it can be used to affect relevant eco-label variables. The purpose is to validate the hypothesis that there is a unique model of feelings/emotions by consumer of eco-products in the pre-purchase stage.

The results from these studies were used in the final design of the consumer product to validate the proposed Green Emotion Model (GEM) 2.0 framework.

### 3.1. Exploratory Survey on Pre-Purchase Emotion

The survey aims to identify the pre-purchase affect from a combination of field and online survey of consumers. Affective models proposed by previous studies were too broad and have a variety of context of application that may not be applicable in the current study of eco-products. It was hypothesized in environmental studies that buying eco-products will engender a unique affective response. Determining a unique set of emotional responses in the pre-purchase stage is critical for the consequent appraisal of the influence of ecological attributes. The objectives of the survey are as follows [49]:

- Determine the participants perception and understanding towards green products.
- Established their willingness to pay for the green features of the product.
- Identify the types of emotions that these products evoke.

Consumption Emotion Set (CES) was used for this survey [50]. There were 62 feelings/emotions shown to the respondents, which they used to illustrate the affect encountered when purchasing an eco-product and non-eco-product. Aside from this, they were probed for specific emotions they have

experienced that were not on the list. It was predicted that the result would yield a smaller set of emotions that would be relevant in the context considered.

### 3.1.1. Participants

A total of 96 respondents participated in the exploratory survey of pre-purchase affect. Majority of them comprising of 82 participants responded through online survey and the remaining 14 answered through a field survey in a selected company that offers eco products in the Philippines.

### 3.1.2. Results of the Survey

The study revealed that 87.80% of the respondents patronize eco-products while 12.20% have not bought any eco-products in the past because they think they are more expensive than alternative products. Participants understandings or knowledge of eco-products can be seen in Table 1 below.

**Table 1.** Environmental Understanding of eco-products.

| Environmental Understanding | Percentage |
|---|---|
| They are not harmful to the environment | 17.91% |
| They are made from organic ingredients without any toxic pesticides and herbicides | 17.66% |
| They are healthy for people, animals and the environment | 16.92% |
| They address recycling, sustainable production and reduction of energy consumption and transport | 16.67% |
| They reduce impacts on the environment | 16.42% |

Source: Gutierrez et al. [49].

A large number of participants experience more positive feelings in purchasing eco-products. The average frequency of participants that experience positive responses is 21.44 as compared to traditional products with 5.38. In contrast, participants experience more negative feelings/emotions when buying traditional product (showing a mean frequency of 5.30) than when buying an eco-product (showing a mean of 1.30) based on their average responses.

Aside from this, a Pareto analysis was conducted to determine whether there's a difference between the two sets of feelings/emotions within each category (eco-products and traditional products). The list of the eighteen (18) feelings/emotion statements that were included in the top 80% is shown in Table 2. It was also observed that though a large number of the participants acknowledge more positive feelings/emotions when buying eco-products, there were still some participants who felt otherwise (i.e., had negative feelings/emotions). The negative feelings/emotions felt by some participants may be due to the suspicion on the green claims on product labels. Some participants also thought eco-products give them a sense of having less quality than non-eco-products. As such, it is worthwhile to investigate whether frequencies of positive and negative emotions differ between these two types of feelings/emotions.

A Kruskal-Wallis test was done to determine the significance of the difference between the average frequencies of positive and negative feelings experienced in patronizing eco-products. A non-parametric test was assumed appropriate since the data violated the assumption of equal variances, thus preventing the use of a t-test or Analysis of Variance (ANOVA). Levine's test for equality of variances was found to be violated at $F(1,59) = 46.997$, $p = 0.00$. The Kruskal Wallis test revealed that there was a statistically significant difference between the type of affect experienced in purchasing a traditional product ($H(1) = 38.709$, $p = 0.000$), with an average rank of 46.59 for positive feelings/emotions and 18.24 for negative feelings/emotions. This shows that there is enough statistical evidence to conclude that the participants experienced more positive feelings/emotions than negative feelings/emotions when patronizing green product [49].

**Table 2.** List of the Top 18 Emotions when Buying an Eco-product.

| Statement | Frequency | Cumulative Frequency | Cumulative % |
|---|---|---|---|
| Good | 49 | 49 | 7.47% |
| Happy | 49 | 98 | 14.94% |
| Optimistic | 49 | 147 | 22.41% |
| Glad | 34 | 181 | 27.59% |
| Pleased | 34 | 215 | 32.77% |
| Hopeful | 33 | 248 | 37.80% |
| Encourage | 31 | 279 | 42.53% |
| Amazed | 29 | 308 | 40.95% |
| Calm | 29 | 337 | 51.37% |
| Caring | 25 | 362 | 55.18% |
| Concerned | 25 | 387 | 58.99% |
| Excited | 25 | 412 | 62.80% |
| Delighted | 24 | 436 | 66.46% |
| Contented | 22 | 458 | 69.82% |
| Peaceful | 21 | 479 | 73.02% |
| Enthusiastic | 20 | 499 | 76.07% |
| Fulfilled | 20 | 519 | 79.12% |
| Compassionate | 17 | 536 | 81.71% |

Source: Gutierrez et al. [49].

*3.2. Eco-Label Awareness*

This part wants to determine the knowledge of consumers towards the eco-labels found on personal care products. The results of the study will be used in the final design of consumer products for a laboratory experiment to validate the proposed framework.

3.2.1. Participants

A total of 100 participants answered the online survey. The majority were female (58%) and males accounted for 42% of respondents. The mean age of the participants was 30.65 years old with a standard deviation of 9.26. The majority of the participants were professionals, accounting for 49% of respondents, college students comprised 43% and the remainder were housewives (8%).

3.2.2. Top Consumer Products

From the results, it was found that the top 3 consumer products bought were shampoo (100%), facial wash (92%) and bar soap (89%), respectively. They used them every day and most participants buy them every three weeks (64%).

3.2.3. Factors Noticed in Packaging

Respondents rated the factors using a 5-point Likert scale from 1 (least aware) to 5 (most aware) when buying consumer products. The results showed that brand (5.22), product quality (4.33) and features of the product (4.02) were the things they notice when buying products. The environmental seal/label on the product got the lowest rating (1.67).

3.2.4. Eco-Labels

The last part of the survey showed the participants 13 Type 1 eco-labels. The eco-labels presented used a scale of 1–10: from less aware with a score of 1 to most aware with a score of 10. These 13 eco-labels are common labels found in consumer products in Asia. The outcome of the survey showed that the top eco-label was Green Choice Philippines (4.21), since consumers believe that it was developed or endorsed by the government, meaning trust was established. The second was the Leaping bunny (2.83) because consumers were familiar with this label and have seen it on many consumer products to

indicate that no animal testing was done. These top two eco-labels (Green Choice Philippines and Leaping bunny) was used in the next stage of the study, which is to validate the proposed framework.

### 3.3. Eco Emotion Structure

The shortlist of emotions from the first part of the study was still numerous and not suitable for use as a measurement tool in future studies of eco-products. Thus, there remained a need to know the shortlist's factor structure and identify redundancies. This part of the study used the shortlisted emotions to measure affective experience in an actual experiment that employed ecological products in the purchasing context.

### 3.3.1. Participants

A total of 100 participants were used to analyze the factor structure of emotions collected from the first study. College students and professionals that were 18 years old and older were invited to join. They were chosen based on their own experience in buying their personal care products. A total of 63% of the total participants were female and the remaining 37% were male. The ages ranged from 18 to 25 years old (44%) and most participants were professionals (87%). Lastly, 75% of the participants were college graduates.

### 3.3.2. Method

Subjects were asked to imagine that they are inside a grocery store to buy shampoo. Shampoo was used since in the eco-label awareness portion of the experiment, it was found to be the top personal care product bought by consumers. They were shown different brands of shampoo during the process. A total of eight shampoos of different brands (Dove, Head, and Shoulder, Pantene, Rejoice, Human Nature, Sunsilk and Palmolive) were shown to the participants. These products were locally available and consisted of eco-friendly and non-eco-friendly brands. Both types of shampoos (eco and non-eco) were shown in order to avoid bias towards eco-friendly products.

Respondents were asked to choose a shampoo they want to purchase. The list of emotions generated from the previous survey was shown to the participants and each of them was rated on a 5-point Likert scale for the experience of the participant with 1 = not at all felt and 5 = very much felt.

### 3.3.3. Statistical Analysis

A factor structure was derived from the data based on the ratings respondents gave for the different emotions using Factor analysis. Factor analysis is an independence technique that aims to define the underlying structure among the variables. This study aimed to determine the emotional reactions of participants towards eco-products. The ratings obtained were used to determine the correlation of the emotions and through factor analysis, a smaller set of emotions was derived. Factor analysis was conducted using the Minitab software. The principal component method was used for extracting factors and varimax was used for rotation. Table 3 below contains the unrotated factor loadings and commonalities.

Unrotated factor loadings show the factor loadings for each variable. Based on this factor loading, the variables glad, delighted, encouraged, enthusiastic, optimistic, passionate, hopeful, good, fulfilled, pleased and happy can be grouped into one variable and classified as glad. Factor 2 can also be grouped and can be classified as peaceful. By looking at the % of variance, it can be inferred that factor 1 accounts for 65.1% of the variability in all 18 variables (emotions/feelings statements), while factor 2 accounts for 4%.

The same analysis was conducted on the data set to get the rotated factor loadings. The results are shown in Table 4 below.

Rotated factor loadings show the factor loadings for each variable. Based on these factor loadings, fulfilled, encouraged, contented, glad and delighted can be grouped into one variable and categorized as contented. The factor 2 loadings peaceful, optimistic and hopeful can be categorized as hopeful.

These two-factor loadings of contented and hopeful are used in the succeeding study to validate the framework. These two generated emotions are included in the survey. These emotions are now applicable for buying eco-products, particularly for personal care products in the condition of the pre-purchase stage.

The % of variance revealed that factor 1 accounts for 34.6% of the variability in all 18 emotions while factor 2 accounts for 34.5%.

**Table 3.** Unrotated Factor Loadings and Communalities.

| Variable | Factor 1 | Factor 2 | Communality |
|----------|----------|----------|-------------|
| Amazed | 0.694 | 0.060 | 0.486 |
| Calm | 0.681 | −0.125 | 0.479 |
| Caring | 0.709 | −0.202 | 0.544 |
| Concerned | 0.729 | 0.095 | 0.541 |
| Contented | 0.783 | 0.334 | 0.725 |
| Delighted | 0.865 * | 0.133 | 0.766 |
| Encouraged | 0.858 * | 0.272 | 0.811 |
| Enthusiastic | 0.858 * | 0.104 | 0.747 |
| Excited | 0.806 | −0.022 | 0.650 |
| Fulfilled | 0.821 | 0.338 | 0.788 |
| Glad | 0.870 * | 0.142 | 0.777 |
| Good | 0.830 * | −0.062 | 0.693 |
| Happy | 0.814 * | −0.058 | 0.665 |
| Hopeful | 0.841 * | −0.210 | 0.751 |
| Optimistic | 0.850 * | −0.179 | 0.755 |
| Passionate | 0.850 * | −0.132 | 0.739 |
| Peaceful | 0.809 | −0.440 | 0.848 |
| Pleased | 0.819 * | −0.074 | 0.676 |
| Variance | 11.724 | 0.717 | 12.442 |
| % Variance | 0.651 | 0.040 | 0.691 |

\* Significant results.

**Table 4.** Rotated Factor Loadings and Communalities.

| Variable | Factor 1 | Factor 2 | Communality |
|----------|----------|----------|-------------|
| Amazed | 0.534 | −0.448 | 0.486 |
| Calm | 0.394 | −0.570 | 0.479 |
| Caring | 0.359 | −0.644 | 0.544 |
| Concerned | 0.583 | −0.448 | 0.541 |
| Contented | 0.790 * | −0.317 | 0.725 |
| Delighted | 0.706 * | −0.517 | 0.766 |
| Encouraged | 0.799 * | −0.414 | 0.811 |
| Enthusiastic | 0.681 | −0.533 | 0.747 |
| Excited | 0.554 | −0.585 | 0.650 |
| Fulfilled | 0.820 * | −0.342 | 0.788 |
| Glad | 0.715 * | −0.515 | 0.777 |
| Good | 0.544 | −0.630 | 0.693 |
| Happy | 0.534 | −0.616 | 0.665 |
| Hopeful | 0.447 | **−0.742** | 0.751 |
| Optimistic | 0.475 | **−0.728** | 0.755 |
| Passionate | 0.508 | −0.694 | 0.739 |
| Peaceful | 0.261 | **−0.883** | 0.848 |
| Pleased | 0.528 | −0.631 | 0.676 |
| Variance | 6.2347 | 6.2071 | 12.4418 |
| % Variance | 0.346 | 0.345 | 0.691 |

\* Significant results.

*3.4. Learnings from Pilot Testing*

The main purpose of a pilot test is to ensure that the experiment meets the researcher's expectations in terms of the information that will be gathered and in terms of the experimentation design [48]. One of the purposes of a pilot study is to determine what are the flaws of the methodology in the study. In pilot testing, problems will be detected, and improvements can be implemented for better experimentation.

Concerning the eye tracker, it is important to have an accurate calibration of the marker detector. The participant should be given clear instructions to look at the markers one by one and cooperation is the key to the success of this process. Also, the participant needs to minimize their head movement during the experiment.

For the pre-testing of the survey, suggestions and comments from the participants were incorporated and some questions were paraphrased to eliminate misinterpretation.

## 4. Validation of the Framework

This part of the study aims to validate the framework showing that environmental attitudes and awareness of consumers are crucial for them to inspect and notice eco-information and thus elicit emotion. These environmental labels should be able to capture the attention of consumers and provoke positive emotions. Affective quality can then influence consumer desires to purchase the products with those labels.

*4.1. Actual Laboratory Experiment*

The validation of the proposed framework was done in De La Salle University laboratory Science and Techology Research Center (STRC) room 218 because a Dikablis eye tracker was housed there. Aside from this, the location was also chosen to minimize the effects of noise factors present in the surroundings. This choice was also made to increase the accuracy of the survey in terms of capturing the perceptions of the participant towards eco-products.

4.1.1. Profile of Participants

A total of 100 participants were gathered and each gender was equally represented with 50 males and 50 females. The age range of the participants ranged from 18 to 62 years old with an average of 22.49 and a standard deviation of 7.59. The majority of them were students (88%) while 11% were professionals and 1% were housewives. As for educational attainment, 73% were high school graduates since most of them were in their 3rd and 4th year in college, 23% had a college degree and 4% had post-graduate degrees.

4.1.2. The Mental Model of Eco-Product/Eco-Label

Participants were asked to draw images of an eco-product/eco-label and they were told they could draw as many images as they like. A total of 113 drawings were leaf/leaves, 43 drew the recycling symbol (♻) and 41 drew a tree. Based on the drawings, participants associate eco-labels with aspects of nature such as leaves and trees. As for the recycling symbol, it is a common symbol found in most eco-products and therefore, they were familiar with it.

4.1.3. Visual Attention Using the Dikablis Eye Tracker and D-Lab Software

Participants were asked to use the Dikablis eye tracker device available in the laboratory. Shampoo was used for this study since the result of the exploratory survey and eco-label awareness study showed that it was the top consumer product bought. For this study, each shampoo design was presented to the participant. There was a total of 16 non-generic brand shampoos with varied placement, color and font size of the eco-labels presented to the participants. The objective was to determine whether they noticed the eco-label on the packaging. We wanted to know if the eco-label was able to capture the visual attention of the participants. The total glances, number of glances

and mean glances were recorded for each participant. Total glances are the accumulated duration of glances towards the area of interest for the selected time interval. Several glances refer to the area of interest for the selected time interval. Lastly, the mean glance is the average duration of glancing at the area of interest for the selected time interval. The area of interest is the eco-label placed on the shampoo. A total of 48 participants used the eye tracking device since the eye tracker failed to function afterwards. At the end of the experiment, the participants were asked to choose one shampoo that they will buy and the reason for buying them. Data collected from this stage were analyzed using structural equation modeling (SEM), which will be discussed in the following section.

### 4.1.4. Survey Questionnaire

During the laboratory experiment, participants answered the survey using Google Docs to determine their environmental attitude and understanding, the emotions experienced while inspecting the product and their desirability and purchase intention. The major findings in the survey are presented below. The data collected from this survey were also included in the structural equation modeling (SEM).

### 4.1.5. Environmental Attitudes and Understanding

The purpose of this experiment was to validate the model showing that environmental awareness and understanding will lead consumers to notice eco-labels. The survey uses the 5-point Likert scale ranging from 1 (not understood) to 5 (extremely well understood). Based on the results of the survey, participants have a high understanding of eco-products with an understanding above 4 for all attributes. Among all the attributes, participants understand that eco-products are not harmful to the environment. On the other hand, participants have the least understanding of eco-products as products that tackles the issues of recycling, sustainable production and reduction of energy consumption and transport. Overall, the understanding of the participants of eco-products was found to be at an average of 4.3467. This means that the participants have a strong understanding of eco-products. In terms of familiarity, the responses of the participants to eco-labels were equally divided with 50% being familiar and 50% not being familiar.

Participants were inquired to rate their attitudes on a semantic scale with regards to buying the eco-product instead of the non-eco-product. The study revealed that the majority of the responses were on the positive attitude side. For them, buying an eco-product is good, with a majority (65%) giving it a rating of 5. Also, more than half of them (54%) believed that it is appropriate to buy an eco-product and gave this a rating of 5. Sixty-one percent said that it is right to patronize stores with eco-products.

### 4.1.6. Emotion Quality

Participants were asked about the emotions they experienced while inspecting the shampoos using a 5-point Likert scale ranging from 1 (not at all) to 5 (very much). The emotions presented to the participants were amazed, cheerful, contented and hopeful. The results of the survey were as follows: 86% of the participant felt a bit amazed to moderately amazed, 85% of the participants were moderately and very cheerful, 92% of the participants were moderately to very content, and 88% of the participants were moderately to very hopeful upon seeing the products.

### 4.1.7. Desirability and Purchase Intention

Participants were asked for the desirability of buying a product/shampoo presented to them as well as their purchase intention. A 5-point Likert scale was used with a range of 1—extremely false/weak to 5—extremely true/strong. Participants were asked to rate their desire to buy the product presented and 79% felt a strong desire towards buying the product, with an average of 3.98 as the strength of desire towards the product chosen. Participants were asked to rate their purchase intention towards the product chosen and 83% felt a strong purchase intention towards the product, with an average of 3.99 as the strength of the purchase intention.

### 4.1.8. Debriefing the Participants

Participants were asked to choose a shampoo they will buy from among the 16 shampoos they evaluated. They were also asked about their motives for choosing that shampoo. In addition, they were asked about their comments on the experiment.

### *4.2. Statistical Analysis*

SEM is a statistical tool used by biologists, economists, educational scholars, marketers, medical researchers and a variety of social and behavioral scientists [51] SEM has been widely used in environmental sustainability studies or pro-environmental behavior studies, as well as in a variety of fields including environmental psychology [52].One advantage of SEM is that one latent variable can be a dependent variable in one set of relationships, and at the same time can be an independent variable in another set of relationships. SEM was used in this study to validate the Green Emotion Model (GEM) 2.0 for analysis.

In this model, the latent variables that cannot be directly measured or observed are environmental understanding and awareness, affective/emotion and purchase intention. The measurement variables for affective quality are the ratings of feeling amazed, cheerful, contented and hopeful. The exogenous variables are the eco-label/logotype, color, size, description and location. In regard to the endogenous variables, total glances, the number of glances and the mean number of glances were considered for visual attention. The sample size used was 768, since 16 shampoos were evaluated for the 48 participants.

### 4.2.1. Independent Variable

### Eco-Information

Experiment labels were designed using $2^4$ factorial designs because there are four factors (the location, color, size and type of eco-label used), so a total of 16 runs/shampoo designs was presented to the participants. The eco-label was either a Green Choice Philippines (GCP) label, which is the only Type 1 eco-label in the Philippines, or the Leaping bunny, since this is the most common eco-label found on personal care products.

### 4.2.2. Dependent Variable

### Visual Attention

The objective of this stage of the experiment was to determine whether respondents noticed the eco-label on product packaging. We wanted to know if the eco-label could able capture the visual attention of the experiment participants. Information regarding the mean number of glances, total number of glances and the local glances of participants towards the eco-label was measured using the eye tracker.

### Emotions

Participants were asked about the emotions they experienced while inspecting the shampoos using a 5-point Likert scale.

### Purchase Intention

Participants were asked about the desirability of buying the shampoo presented to them as well as their purchase intention. A 5-point Likert scale was used ranging from 1 (extremely false/weak) to 5 (extremely true/strong).

### 4.2.3. Mediating Variable

Environmental Attitude

The purpose of this part of the experiment was to validate the model by showing that environmental awareness and understanding will lead consumers to notice eco-labels. People who have greater concern for the environment will more highly value or prefer eco-products in the marketplace. This correlation was used as a mediating variable, since the attitude of the participants towards eco-products can affect their emotional responses and intention to read a label.

### 4.3. Data Analysis Using Structural Equation Modeling (SEM)

Before the actual integration as a SEM model, the model's validity and reliability were examined to avoid plausible errors from arising in the computation of regression weights refer to Table 5. Composite Reliability (CR), Average Variance Extracted (AVE), Maximum Shared Variance (MSM) and Average Shared Variance were found to be viable measures based on their validity and reliability. All of the CR is above the threshold value of 0.7 (higher the better), which means the data set under the factor was reliable. The AVE, which is recommended to be above 0.5, means no convergent validity issues arose. MSV and ASV were less than AVE, which means no discrimination validity arose. In general, the fit indices indicated that the compact model was well-fitted, as can be seen in Table 5 below. Given this, the data was now ready to be inputted as a SEM model.

**Table 5.** Validity and Reliability Computation.

| Regression Weight | CR | AVE | MSV | ASV | Understanding | Emotion | Awareness | Intention |
|---|---|---|---|---|---|---|---|---|
| Understanding | 0.704 | 0.784 | 0.212 | 0.082 | 0.674 | | | |
| Emotion | 0.936 | 0.833 | 0.187 | 0.063 | 0.014 | 0.913 | | |
| Attitude | 0.807 | 0.591 | 0.212 | 0.089 | 0.460 | −0.051 | 0.769 | |
| Intention | 0.777 | 0.657 | 0.187 | 0.091 | 0.185 | 0.432 | 0.231 | 0.811 |

Data collected from the results of the Dikablis eye tracker and survey were analyzed using SEM. The basic steps in SEM were followed according to psychologist Paul Kline [53]. First, a conceptual model representing the hypotheses on the relationship between eco-information attributes, environmental attitude, visual attention, emotion and purchase intention was created. The proposed GEM 2.0 presented in the previous discussion was translated into a structural equation model, as shown in Figure 2.

The model in Figure 2 (hereafter referred to as the "imputed model") was formulated using the automatic formulation macro in SPSS AMOS ver. 23 (IBM, Armonk, NY, USA). The imputed model formulated was imputed, meaning all variables were connected on a common latent factor (CLF) to account for possible errors in the data gathering method. After this, the model fit was analyzed in the same way as the CFA with metrics to be considered. The imputed model deals with smaller measurements, which can mean that the dataset could be more easily fitted. However, this does not necessarily mean that the automatic formulation macro is more correct; the sensitivity of the result is the basis for making such a decision on model correctness.

In general, the fit indices shown above indicated that the imputed model is good. Moreover, the imputed model had enhanced parsimony, sensitivity, and a much-improved model fit. Therefore, it was decided that the analysis would begin to employ the imputed model. The final SEM model is shown in Figure 3 with the path coefficients.

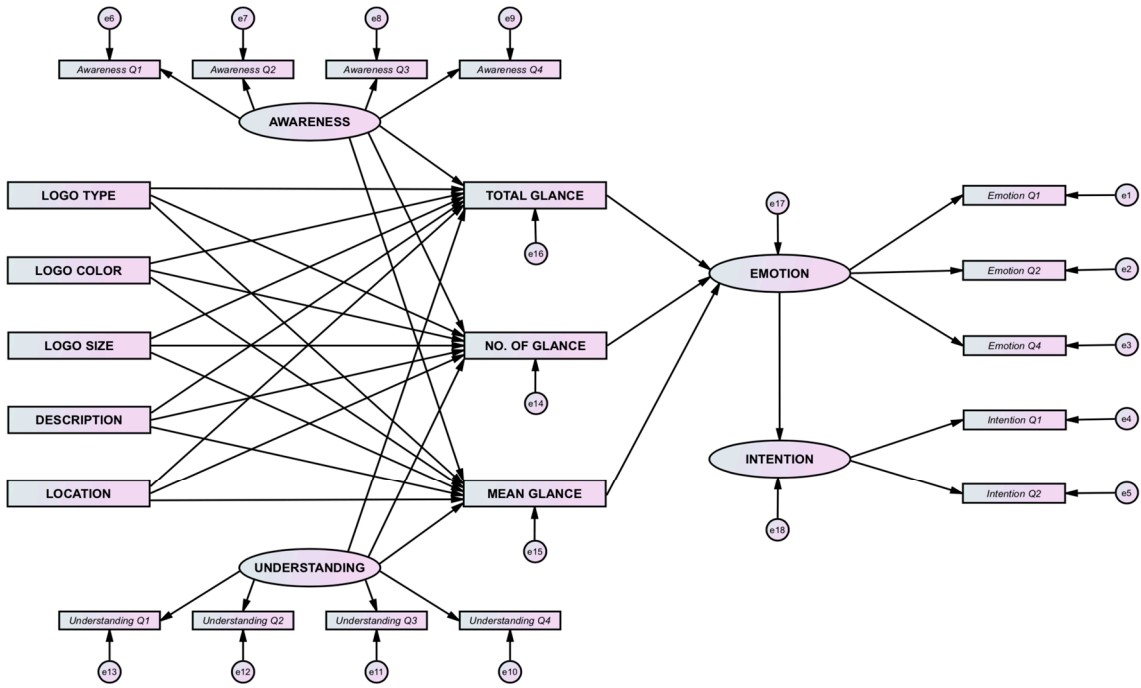

**Figure 2.** The initial Structural Equation Model.

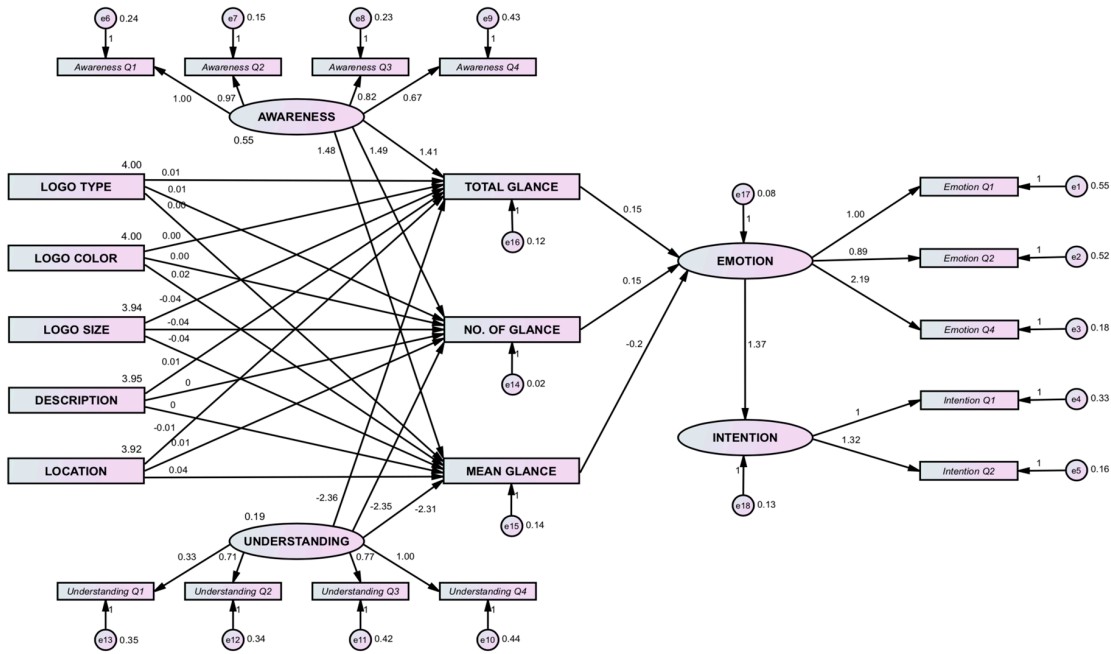

**Figure 3.** Final SEM Model.

*4.4. Final Structural Equation Model*

Table 6 shows the path coefficient, standard error and standardized estimate for the model. Significant results were highlighted with an asterisk. As can be seen, environmental attitude and awareness are significant variables in the models for predicting consumer attention. Consumer emotions also predict consumer purchase intention, particularly for more intensely felt emotions.

**Table 6.** Path Coefficient of the Structural Model.

| Path | | Path Coefficient | Std. Error |
|---|---|---|---|
| Environmental Attitude | Glance Total | 1.406 * | 0.103 |
| Environmental Attitude | Number of Glances | 1.493 * | 0.101 |
| Environmental Attitude | Mean Glances | 1.48 * | 0.105 |
| Environmental Understanding | Glance Total | 2.36 * | 0.273 |
| Environmental Understanding | Number of Glances | −2.352 * | 0.268 |
| Environmental Understanding | Mean Glances | −2.306 | 0.269 |
| Logotype | Glance Total | 0.007 | 0.029 |
| Logotype | Number of Glances | 0.007 | 0.027 |
| Logotype | Mean Glances | 0.002 | 0.029 |
| Logo color | Glance Total | 0.001 | 0.029 |
| Logo color | Number of Glances | 0.002 | 0.027 |
| Logo color | Mean Glances | 0.015 | 0.029 |
| Logo size | Glance Total | −0.042 | 0.029 |
| Logo size | Number of Glances | −0.044 | 0.028 |
| Logo size | Mean Glances | −0.039 | 0.03 |
| Logo description | Glance Total | 0.011 | 0.029 |
| Logo description | Number of Glances | −0.001 | 0.028 |
| Logo description | Mean Glances | −0.002 | 0.03 |
| Logo location | Glance Total | −0.007 | 0.03 |
| Logo location | Number of Glances | 0.014 | 0.028 |
| Logo location | Mean Glances | 0.037 | 0.03 |
| Glance Total | Emotion | 0.154 | 0.064 |
| Number of Glances | Emotion | 0.149 | 0.08 |
| Mean Glances | Emotion | −0.204 | 0.065 |
| Emotion | Intention | 1.372 * | 0.273 |
| Emotion | Emotion 1 | 1 | |
| Emotion | Emotion 2 | 0.886 * | 0.215 |
| Emotion | Emotion 4 | 2.193 * | 0.394 |
| Intention | Purchase Intention 1 | 1 | |
| Intention | Purchase Intention 2 | 1.324 * | 0.139 |
| Environmental Attitude | Environmental Attitude 1 | 1 | |
| Environmental Attitude | Environmental Attitude 2 | 0.97 * | 0.061 |
| Environmental Attitude | Environmental Attitude 3 | 0.82 * | 0.061 |
| Environmental Attitude | Environmental Attitude 4 | 0.674 | 0.07 |
| Environmental Understanding | Environmental Understanding 6 | 1 | |
| Environmental Understanding | Environmental Understanding 5 | 0.768 * | 0.136 |
| Environmental Understanding | Environmental Understanding 3 | 0.715 * | 0.123 |
| Environmental Understanding | Environmental Understanding 1 | 0.334 | 0.103 |

$* p < 0.05$.

## 5. Discussion

The results obtained from the path analysis showed that environmental attitude is a significant predictor of the glance total, number of glances and mean glance rate ($p < 0.005$). The path coefficients show that environmental attitude increases visual attention measurements by as much as 140%.

Environmental attitudes were captured by four measures in this study. The first one (Environmental attitude 1) is the general attitude towards buying eco-products. The majority of respondents (65%) had a positive general attitude. The second measure is the appropriateness of buying eco-products (Environmental attitude 2). More than half (54%) of respondents believe that it is appropriate to buy eco-products and sixty-one percent said that it is morally right to patronize eco-product stores (Environmental attitude 3). Overall, the results showed that participants have a positive attitude towards eco-labels. This suggests that environmental attitudes play a vital role to encourage consumers to notice messages on an eco-label.

Those that care for the environment have a natural tendency to look at the content of eco-labels because this content helps them make the right choices when shopping.

The eco-label provides the consumer with visual information they need as they examine a product, meaning these labels are crucial in consumer decisions to buy that product. A consumer with a positive environmental attitude will look for the eco-information on a product. This highlights the importance of well-designed labeling because it significantly impacts the consumers' perceptions of the ecological friendliness of that product. It also shows the importance of improving the design of an eco-label so that consumers can notice and understand the eco-information it presents.

The findings also validated that environmental understanding is a significant predictor of the glance total, number of glances and mean glances rate ($p < 0.05$). However, it can be seen that the coefficients for this relationship are negative. While there is a common belief that those who understand concerns about the environment will pay more attention to eco-labels, this was not found to be true in the study. Participants that do not understand environmental concerns are more likely to look closely at eco-labels so they can better understand the information they provide.

Since most participants did not understand the eco-labels we provided, they looked at the labels to understand and decipher them. Participants who were not familiar with environmental issues looked at the eco-labels for 236% more time than other participants. Environmental knowledge provided by eco-labels is important to help consumers understand environmental issues. Environmental knowledge can also trigger concerns and attention among individuals that can lead to an increased willingness to help address environmental problems [31]. Those consumers who have feelings of environmental altruism understand and thus notice eco-labels.

Emotions also revealed to be a significant predictor of the glance total and mean glances rate. This validates the claim that when the participants noticed eco-labels, the labels elicited emotions. These emotions also showed significance in predicting consumer purchase intention ($p < 0.05$), thus validating the hypothesis that emotion stimulates consumers to purchase eco-products. Based on these results, the intensity of emotion increases the purchase intention by 137%. Purchase intention represents the willingness of consumers to buy and desire to have a product. This finding validates previous studies that suggest that consumers relate feelings of having a good conscience and feelings of responsibility for the well-being of one's family with organic food purchasing decisions [16,46–48]. It can be said that consumers attach a higher emotional value to green products. The consumer who favors going green as an act to help protect the environment has the potential to experience positive feelings from the perception of completing a good deed for themselves and for society at large. These findings validate the hypothesis that a positive environmental attitude strengthens the relationship between ecological attributes and emotions.

The emotions of being amazed, cheerful and hopeful have good loadings for the overall construction of the emotion's variable, as can be seen by the factor loadings.

The study revealed that the ecological attributes focused on in this study are not the ones that most significantly affect the visual perception of participants. The participants may have had preconceived notions that affected their evaluations of the shampoos presented to them. However, based on an interview with participants, the participants did notice the eco-labels. The majority preferred the eco-labels to be located at the center of the shampoo packaging. They also noticed colored eco-labels much more often compared to black and white (monochrome) labels. Concerning the type of eco-label, respondent attention was captured more by the Green Choice Philippines than by the Leaping Bunny eco-label. The study revealed that participants trust the Green Choice Philippines eco-label because they believe it is endorsed by an independent organization. The success of eco-products depends on consumers' trust in the eco-information that accompanies these products. Consumers will only prefer eco-labels which they consider as being trustworthy. This also suggests that regulated or government-sponsored labels are generally favored over other labels. This study revealed the vital role of eco-labels in affecting the social and behavioral aspects of consumers.

## 6. Conclusions and Recommendations for Future Research

The new GEM 2.0 conceptual framework was proposed to include the importance of environmental attitudes, visual perception and emotion for consumers to decide to purchase eco-products. This model is unique as it is the first to predict emotional experience from ecological attributes and relates consumers' environmental attitudes that encourage them to read the messages of an eco-label to their purchase intention. GEM 2.0 proposed that eco-labels play an important role in encouraging consumers to purchase green products. Designers of eco-labels should consider how consumers look for this environmental information, and this priority must capture their attention. Green marketing should understand the significance of eco-labeling, since it focused on the reputation of achieving and maintaining a green product standing and creating an effective competitive advantage. Eco-labels can be used to indicate the product's influence and as a product differentiator.

The result of this research can be used by companies to identify the environmental and emotional benefits of their green products. By showing consumers the emotional benefits of eco-products, more of these products may be seen in the market, thereby improving the global environment and well-being.

This study has useful lessons for green marketing, since it shows the affective benefit for consumers to successfully position a green product as providing personal care. This research suggests that for a green product to be successful, its emotional benefits should also be conveyed to consumers to attract them to purchase these products.

This study can offer precious knowledge that can help both private and public enterprises to develop and design appealing eco-labels that will be understood and favored by consumers.

Due to the limitation of needing to use the eye tracker equipment that was housed in the Science and Technology Research Center (STRC) 218 laboratory at De La Salle University, the participants were limited to college students, staff and faculty from the university. Future studies can include other participants from different universities and industries in the Philippines to be more representative of the sentiment of consumers regarding eco-labels.

Cross-cultural studies can also be carried out to see if there are cultural differences in consumer acceptance and perceptions towards eco-labels in different countries. Replicating this study in other parts of the world to identify the emotions elicited by these eco-products in a specific population would be an interesting approach for future researchers. The model can also be validated by using other eco-products aside from personal care products.

**Author Contributions:** A.M.J.G. carried out a review of the literature and development of the (GEM) 2.0 conceptual framework, established the framework methodology, carried out data gathering, validated the model and wrote the final paper. A.S.F.C. provided supervision and reviewed and edited the paper. R.S. mentored and provided help in developing the methodology and validating the model. She also helped in the acquisition of funding from the Engineering Research and Development for Technology (ERDT) of the Department of Science and Technology (DOST) in the Philippines. All authors have read and agreed to the published version of the manuscript.

**Funding:** This research was partly funded by the Engineering Research and Development for Technology (ERDT) of the Department of Science and Technology (DOST) in the Philippines.

**Acknowledgments:** I would like to acknowledge the help and support of Ezekiel Bernardo part time faculty of the Industrial Engineering Department of De La Salle University for his expertise on Structural Equation Modelling (SEM) and use of AMOS software.

**Conflicts of Interest:** Engineering Research and Development for Technology (ERDT) of the Department of Science and Technology (DOST) in the Philippines, that funded this research had no role in the conceptualization, design and gathering of data of the study.

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
