# Peer review of "A Proposed Framework on the Affective Design of Eco-Product Labels"

_sustainability, doi:10.3390/su12083234_

Round 1

Reviewer 1 Report

This paper provides an extensive study on the eco-product labels, which is an interesting topic these days. Upon reviewing the manuscript, I would suggest the authors to improve the following:

  1. The introduction, please be concise and straight to the topic. You can justify the topic by mentioning the trend of eco-labelling and the importance of emotions in the purchase decisions.
  2. The literature review needs major rework. Please reduce the irrelevant content and focus on the manuscript's own model development. I would suggest the authors to synthesise the sources, since currently it looks like the descriptions of several theories/models. I would also suggest the authors to develop some hypotheses.
  3. The methodology is ambitious. However, the authors need to specify the methodology. If there is more than one steps, please provide separate subheadlines for each step and please specify the sampling. It would be great to see the instrument/questions and validity-reliability check in this manuscript.
  4. The results are not clearly presented. The decimal rounding is inconsistent and it looks chaotic at the moment, since there is no operational descriptions of the variables. The path coefficient table would be clearer with hypotheses and another column indicating whether the hypothesis is supported/not.
  5. The discussion and conclusions are still on the surface. The author needs to discuss the results and relate them back to the theory. The authors also need to be clear about this paper's contributions, theoretically and practically.
  6. English editing is needed to improve the expression clarity, due to some redundant contents.

Reviewer 2 Report

The topic of the paper, determining the effects of eco-labelling, will be of interest to companies and marketer as well as organizations that seek to promote environmentally friendly consumerism. Knowing what appeals and informs consumers remakes the marketplace into a site of activism and politics for environmental movement. The focus on the environmental attitudes, emotions provoked, visual clues, and product desirability offers a comprehensive picture of what takes place in the consumer-product interaction.  

Since the article's topic and methodological approach is out of my disciplinary area, I will speak mainly to improving the presentation of the paper. Much of the important information about what this article discusses, the innovative framework used, and the key findings is present in the paper but could be better organized. 

-the abstract should more clearly present what is known, what is unknown, and how the study will address gaps in the field. this should include reference to the Green Emotion Model that will be proposed in the paper

-the introduction fell more like a literature review than a concise presentation of what the paper will do: setting the context for the topic, questions, the stakes, the argument, a brief discussion of the methods/data, and an outline of the paper

-the literature was quite lengthy. the disciplinary fields of the paper should be clearly laid out with discussions of the major authors and scholarship. the authors did an excellent job of reviewing previous work and demonstrating the gaps. but this section could be abbreviated if these models are widely known and accepted. 

-the data and methods section sometimes mixed together the experiment and its findings. these should be kept in separate sections (ie. lines 471-475)

-the conclusions could speak about any limitations of the study as well as lay out pathways for future research
